# Identification of New L-Heptanoylphosphatidyl Inositol Pentakisphosphate Derivatives Targeting the Interaction with HIV-1 Gag by Molecular Modelling Studies

**DOI:** 10.3390/ph15101255

**Published:** 2022-10-12

**Authors:** Halilibrahim Ciftci, Belgin Sever, Esra Ayan, Mustafa Can, Hasan DeMirci, Masami Otsuka, Amaç Fatih TuYuN, Hiroshi Tateishi, Mikako Fujita

**Affiliations:** 1Department of Drug Discovery, Science Farm Ltd., Kumamoto 862-0976, Japan; 2Medicinal and Biological Chemistry Science Farm Joint Research Laboratory, Faculty of Life Sciences, Kumamoto University, Kumamoto 862-0973, Japan; 3Department of Molecular Biology and Genetics, Koc University, Istanbul 34450, Turkey; 4Department of Pharmaceutical Chemistry, Faculty of Pharmacy, Anadolu University, Eskisehir 26470, Turkey; 5Department of Engineering Sciences, Faculty of Engineering and Architecture, Izmir Katip Celebi University, Izmir 35620, Turkey; 6Department of Chemistry, Faculty of Science, Istanbul University, Istanbul 34126, Turkey

**Keywords:** HIV, Gag, MA, PI(4,5)P2, IP6, molecular modelling

## Abstract

The HIV-1 Gag protein binds to the host cell membrane and assembles into immature particles. Then, in the course of immature virion budding, activated protease cleaves Gag into its main components: MA, CA, NC, and p6 proteins. The highly basic residues of MA predominantly interact with the acidic head of phosphatidyl-inositol-4,5-bisphosphate (PI(4,5)P2) inserted into the membrane. Our research group developed L-Heptanoylphosphatidyl Inositol Pentakisphosphate (L-HIPPO) and previously confirmed that this compound bound to the MA more strongly than PI(4,5)P2 and inositol hexakisphosphate (IP6) did. Therefore, herein we rationally designed eight new L-HIPPO derivatives based on the fact that the most changeable parts of L-HIPPO were two acyl chains. After that, we employed molecular docking for eight compounds via Maestro software using high-resolution crystal structures of MA in complex with IP6 (PDB IDs: 7E1I, 7E1J, and 7E1K), which were recently elucidated by our research group. The most promising docking scores were obtained with benzene-inserted compounds. Thus, we generated a library containing 213 new aromatic group-inserted L-HIPPO derivatives and performed the same molecular docking procedure. According to the results, we determined the nine new L-HIPPO derivatives most effectively binding to the MA with the most favorable scoring functions and pharmacokinetic properties for further exploration.

## 1. Introduction

Retroviruses are enveloped single-stranded RNA (+) viruses, which include distinct members such as human immunodeficiency virus (HIV), the main causative agent of acquired immunodeficiency syndrome (AIDS). Up to now, several antiretrovirals have been developed that target key processes of the viral life cycle of HIV, delay or prevent the onset of clinical manifestations, and extend the survival of patients with AIDS. According to the mode of action, there are approved drugs against HIV infection and AIDS such as nucleoside reverse transcriptase inhibitors, non-nucleoside reverse transcriptase inhibitors, protease inhibitors, integrase inhibitors, co-receptor inhibitors, and fusion inhibitors [1,2,3,4,5,6,7]. However, some problems are seen in the therapies using these drugs. The biggest HIV problem at the moment is recently-infected cells remain in an HIV-infected patient. The emergence of resistance along with the ability of the virus to evolve by mutation, drug-drug interactions, cotreatment of other AIDS-related diseases, the requirement of compliance with the treatment protocols, and severe side effects also restrict the success of therapy with the current antiretroviral drugs [2,3,4,5,6]. Although relentless efforts have been devoted to finding more effective antiviral drug targets, there is no antiretroviral agent that targets the budding phase driven by the Gag polyprotein precursor (Pr55^Gag^: Gag) [1,2,3,4,5,6,7,8].

Gag attached with a myristoyl group is synthesized, transported to cell membranes, and assembled. After assembly, the immature virion (Gag lattice) buds. Upon budding, the activated viral protease cleaves Gag to form a range of proteins and spacer peptides (SP), namely matrix (MA), capsid (CA), SP1, nucleocapsid (NC), SP2, and p6, which stimulate a series of events for the generation of infectious mature virions (Figure 1) [9,10,11,12,13,14,15,16,17,18,19,20,21,22].

In Gag membrane binding, Gag proteins bind with phospholipids potentially found at the budding site. Phospholipids are substantial components of the plasma membrane and glycerophospholipids contain acidic or neutral polar heads and hydrophobic tails. Gag enables budding through interaction with acidic glycerophospholipids, particularly phosphatidyl-inositol-4,5-bisphosphate (PI(4,5)P2) (Figure 1) in the membrane through MA, a helical Gag domain that exhibits a conserved patch of highly basic residues. PI(4,5)P2 is generally found in large amounts on the cytoplasmic leaflet of the membrane, where Gag mainly assembles in most cell types [23,24,25,26]. Membrane binding is also mediated by the exposure of the *N*-terminal myristic acid moiety that inserts into the lipid bilayer. However, ionic interactions between the acidic polar heads of PI(4,5)P2 with the membrane-facing highly basic residues of MA have been reported to have the edge over the protein *N*-myristoylation on the energetics of in vitro HIV-1 MA membrane binding [27,28,29]. On the other hand, the results of surface plasmon resonance (SPR) analysis performed by our research group revealed that both the divalent phosphate groups and the acyl chains of PI(4,5)P2 are crucial for tight binding to MA. In addition, the dissociation constant (Kd) values diminished with longer acyl chain-substituted PI(4,5)P2, indicating the role of acyl chains in the interaction between MA and PI(4,5)P2. These data also confirmed that a longer carbon chain increased MA affinity [30].

Inositol hexakisphosphate (IP6) (Figure 1) is obtained at the end of a part of an enzymatic cascade initiated by the cleavage of PI(4,5)P2 to form IP3, IP4, and IP5. The position and number of phosphate moieties on the inositol ring create the differences in the members of the phosphoinositide family. As a highly charged polyanion, IP6 was determined to bind to an electropositive pore (bearing a ring of arginine residues), affecting viral production and primary cell replication. IP6 was reported to interact with both the MA and CA domains of Gag. IP6 is released by the proteolytic cleavage of Gag and then stabilizes the mature capsid lattice in the course of maturation [31,32,33,34,35,36,37]. Our research group previously demonstrated that IP6 was able to bind to the MA approximately 10 times more strongly than IP3 [38].

Several strategies have been pursued so far to achieve a cure for HIV-1, such as gene therapy with engineered T cells, immunotherapy with vaccines, and broadly HIV-1 neutralizing monoclonal antibodies [39,40]. However, a “kick and kill” or “shock and kill” approach is the most preferred strategy, which refers to initial reactivation of the viral reservoir with latency-reversing agents (kick) and subsequent elimination of infected cells through virus-mediated cytolysis or immune-mediated clearance (kill). However, the “kill” process requires vaccination in addition to natural host HIV-specific immunity, and there is no effective vaccination option to eliminate latent reservoirs [39,40,41,42]. Therefore, our group proposed a novel approach entitled “lock-in and apoptosis”, in which the compound prevents the budding of offspring virus after the “kick” process and enables the virus to be locked in an HIV-infected host cell, which latterly undergoes apoptosis together with the virus, leading to the complete eradication of HIV from the body. This compound, L-Heptanoylphosphatidyl Inositol Pentakisphosphate (L-HIPPO) (Figure 1), which was designed as an IP6 derivative and synthesized by our group, was determined to be capable of binding to the MA domain of Gag 70-fold more strongly than that of the less phosphorylated PI(4,5)P2 derivative [43].

In the current work, initially, we rationally designed eight new L-HIPPO derivatives related to four new design strategies. After that, we performed molecular docking for new phosphoinositides via Schrödinger software, using three high-resolution crystal structures of the MA in complex with IP6 that our research group had previously identified [44]. The most promising docking scores and interactions were obtained with benzene-inserted compounds. Therefore, we generated a library containing 213 new aromatic group-inserted L-HIPPO derivatives and carried out molecular simulation studies (docking and absorption, distribution, metabolism, and excretion (ADME) calculations) to determine the most promising MA-targeted anti-HIV compounds.

## 2. Results

Our research group carried out the first X-ray free-electron laser (XFEL) study of the interaction of MA with IP6, along with representing the purification, characterization, and microcrystallization of two MA crystal forms acquired in the examination of IP6 [45]. Then, in our recently published paper, we performed both synchrotron cryo X-ray crystallography and ambient-temperature serial femtosecond X-ray crystallography (SFX) with an XFEL to obtain MA-IP6 co-crystal structures. We reported three crystal forms and the resulting X-ray structures of MA in complex with the IP6 molecule (PDB IDs: 7E1I, 7E1J, and 7E1K) (Figure 2) [44].

In light of improved literature and our deep experiences in anti-HIV drug design, we developed four new design strategies (eight new L-HIPPO derivatives (Figure 3)) on the structural modification of two acyl groups to increase the anti-HIV effects of L-HIPPO. The inositol group and acidic phosphates are crucial for binding with the basic pocket of MA. Two acyl chains are also essential for ligation with MA. However, these two acyl groups are the most favorable parts of L-HIPPO for structural modification in order to enhance the binding potential.

These strategies are as follows:Removal of a heptanoyl group to reduce hydrophobicity.Introduction of a fluoro group to increase hydrophobicity.Insertion of a double bond into the heptanoyl group in order to alter the π electron.Insertion of a benzene ring into the heptanoyl group in order to alter the π electron.

The validated molecular docking protocol, with docking scores and binding interactions comparable to the co-crystallized ligands, was applied for eight new L-HIPPO derivatives to explore their binding affinities using three templates, which were recently solved by our group (PDB IDs: 7E1I, 7E1J, and 7E1K) [44].

According to molecular docking results, eight new phosphoinositides (SCH-1, SCH-2, FH-1, FH-2, USH-1, USH-2, BH-1, and BH-2) displayed high affinity in the MA domain (PDB IDs: 7E1I, 7E1J, and 7E1K). These compounds demonstrated the best alignment in the MA domain of 7EI1 (Figure 4). In addition, the docking score with the lowest energy (high negative scores) and stronger interactions was obtained in the MA domain of 7EI1 for all compounds (Table 1). These compounds established key hydrogen bonds and salt bridge formations with key residues such as Arg19, Lys17, Lys25, Lys26, Lys29, and Gln27 (Figure 5A–C). Moreover, the highest docking scores, belonging to BH-1 and BH-2, explained their strongest binding capacities to the MA domain. However, it was observed that another six new compounds possessed lower docking scores compared to L-HIPPO (−7.259 kcal/mol), which were determined to range from −5.303 to −6.268 kcal/mol, and had relatively lower binding potencies. The H-bond and salt bridge formation distances of BH-1 and BH-2 were calculated compared to L-HIPPO, with the angstrom as the unit (Table 2).

The most promising molecular docking results for BH-1 and BH-2 pointed out the importance of the presence of the aromatic moieties at both acyl groups. Therefore, we created a library of 213 new aromatic group-based L-HIPPO derivatives containing distinct substitutions on the benzene and naphthalene rings, suggesting the essential chemical structures of approved drugs and effective compounds, which were searched in the PubChem and ZINC databases containing structural and functional information about different organic compounds. Afterward, we also performed molecular docking studies for new phosphoinositides compared to BH-1 and BH-2 (PDB IDs: 7E1I, 7E1J, and 7E1K). Results demonstrated that nine new L-HIPPO derivatives (H-2, H-6, H-32, H-39, H-62, H-64, H-86, H-189, and H-201) bearing distinct aromatic groups (Figure 6) were found to be more efficient than BH-1 and BH-2 when comparing the docking scores (Table 3). These phosphoinositides were also superimposed using the flexible ligand alignment in Maestro software. The docked compounds demonstrated the best superimposition in the MA binding site of 7EI1 with similar protein–ligand interactions (Figure 7). In general, compounds were embedded in the MA domain through hydrogen bonding, salt bridge formation, and π-cation interactions with Arg19, Glu72, Lys17, Lys25, Lys26, Lys29, Lys31, and Lys94. All these compounds revealed higher affinity when compared with BH-1 and BH-2. Among the new compounds, H-2 and H-6 exhibited the highest docking scores and presented the most important interactions, as depicted in Table 2 and Figure 8A–C. The other compounds were sorted in order of their MA binding potential as H-189 > H-201 > H-39 > H-86 > H-62 > H-64 > H-32.

Some crucial ADME properties of these nine new L-HIPPO derivatives, such as aqueous solubility (CIQPlogS), partition coefficient of brain/blood (QPlogBB), coefficient of central nervous system (CNS), partition coefficient of octanol/water (QPlogPo/w) and compliance to Lipinski’s and Jorgensen’s rules were in silico estimated. The results shown in Table 4 were found plausible related to the specified parameters. The CIQPlogS values of H-2, H-6, H-32, H-39, H-62, H-64, H-86, H-189, and H-201 (−4.571 to −6.432) were found to be within the specified range (−6.5 to 0.5). Additionally, the limits of the QPlogPo/w values of these derivatives were observed as -0.620 to 1.176, which were within the specified range (−2 to 6.5). The QPlogBB values of these derivatives were detected out of the specified limits (−10.299 to −7.647), which were defined as −3 to 1.2. Moreover, the CNS values of these compounds were determined on a −2 (inactive) scale. These phosphoinositides violated three parameters of Lipinski’s rule of five (maximum is four) and one parameter of Jorgensen’s rule of three (maximum is three) (QikProp, Schrödinger, LLC, NY, 2016).

## 3. Discussion

Up until now, there has been no effective cure for HIV/AIDS. In general, long-term multi-drug regimens have to be administered to HIV-positive patients in order to reduce recurrence rates. Despite the great advantages of combination therapy, including synergistic effects, lower toxicity, and lower drug resistance development, remaining latent reservoirs, occurrence of cross-resistance mutations, and the requirement of proper and long-time drug use still prevent the achievement of a permanent cure for HIV. Therefore, there is an urgent need for new therapeutics to be developed to stop the budding stage of the HIV cell cycle in order to effect HIV eradication. At this point, it is known that Gag is essential for potent assembly and budding, and the MA protein of Gag is the main component for its potency in anchoring to the host cell plasma membrane. In this binding mode, electrostatic interactions between highly basic residues of MA with negatively charged PI(4,5)P2 of the membrane serve preferential functions for protein–membrane interactions.

Our group previously screened several phosphoinositides bearing a different number of phosphates and their regioisomers with or without an acyl chain, using an SPR assay for the determination of *K*d values for the binding of MA, and found that both phosphate groups and acyl chains were required for strong binding with MA [30]. Our group also demonstrated that IP6 was capable of binding to the MA domain, notably when compared to the less phosphorylated IP3, indicating the importance of the presence of more phosphate groups in the inositol ring [38]. Our first report of XFEL study for the determination of MA and IP6 interaction revealed that SFX is suitable for the diffraction of microcrystal forms of the MA-IP6 complex, which were diffracted to beyond 3.5 Å resolution [45]. Our recently published study clarified the MA structures in complex with IP6 (PDB IDs: 7E1I, 7E1J, and 7E1K) and the interactions between them were orchestrated by hydrogen bonding with basic residues Lys and Arg together [44].

Our research group designed L-HIPPO based on the fact that the MA domain of Gag mediates membrane binding through its interaction with PI(4,5)P2 in the membrane and this phosphoinositide derivative was determined to possess robust MA-binding affinity. Structurally, L-HIPPO is a diheptanoylglycerol ester of IP6 [43]. Due to the fact that these ester groups were crucial to interaction with MA, modification of the diheptanoylglycerol moiety would enhance the MA binding of L-HIPPO.

Molecular docking studies were performed for eight new L-HIPPO derivatives to gain insight into the interactions between protein structures and ligands, which can further explain the ligand specificities and differences in the binding site of the target structure (MA domain) using our three recently released templates (PDB IDs: 7E1I, 7E1J, and 7E1K). The docking scores, which involve a summary of glide scores obtained from the standard-precision or extra-precision scoring functions, enabled us to make comparisons between different ligands, whereas the emodel scores were suitable for comparing the different conformations of the same ligand [46]. According to the docking outcomes, the highest docking scores were attained with benzene ring substitution, implying that the presence of aromatic groups in the L-HIPPO structure augmented MA binding affinity. Although the aromatic groups played no role in direct interactions, these groups changed the space that was occupied by the total ligand, leading to stronger interactions. The docking scores of another six compounds were found to be lower than L-HIPPO. The distances of H-bond and salt bridge formation for BH-1 and BH-2 were found to be greater than that of L-HIPPO. This outcome indicated that shorter H-bond and salt bridge formation distances between the phosphates of L-HIPPO with highly basic residues led to the participation of a molecule in a narrower space, thus the acyl chains were also embedded in a narrower hydrophobic region compared to BH-1 and BH-2. On the contrary, the settlement of BH-1 and BH-2 in MA with greater H-bond and salt bridge formation distances of the inositol ring and basic residues also led to the increase of occupation of acyl chains in a larger hydrophobic space in MA, enhancing the binding affinity of BH-1 and BH-2 in MA [47,48,49,50,51,52,53]. These findings encouraged us to create a large library containing 213 new L-HIPPO derivatives containing distinct aromatic groups. Then, the molecular docking procedures were also employed for these new derivatives compared to BH-1 and BH-2 (PDB IDs: 7E1I, 7E1J, and 7E1K). Among a large number of compounds, nine new compounds (H-2, H-6, H-32, H-62, H-64, H-86, H-189, and H-201) showed more significant binding affinity compared to BH-1 and BH-2. The general chemical framework of the most effective derivatives established that substitutions such as hydroxy, methoxy, cyano, chloro, amino, and trifluoromethyl on the phenyl ring instead of methyl, and the exchange of p-tolyl to naphthyl or benzothiazole rings, enhanced the binding affinity of these compounds compared to p-tolyl-substituted BH-1 and BH-2. Furthermore, H-2 and H-6 were defined as the most potent phosphoinositides for MA binding. It can also be deduced that 3,4-dihydroxyphenyl and 3-methoxy-4-hydroxyphenyl substitutions played important roles for the high binding orientations of H-2 and H-6, respectively. In particular, the 3,4-dihydroxyphenyl moiety of H-2 presented hydrogen bonding with Glu72, while the 3-methoxy-4-hydroxyphenyl moiety of H-6 was able to form key hydrogen bonding with Lys17. This outcome pointed out that the substitutions in the acyl chains of H-2 and H-6 contributed to the hydrogen bonding capacity of these compounds. The stronger affinity of H-2 and H-6 could render the higher MA binding affinities of these compounds compared to other tested phosphoinositides.

The pharmacokinetic properties of anti-HIV drugs have a great importance in the success of the therapy. The balance of the hydrophilicity and lipophilicity of an anti-HIV drug is essential because this drug has to be principally present in aqueous environments, including blood serum as HIV infects CD4^+^ leucocytes, and this drug has to cross the lipid plasma membrane in order to enact its pharmacological effects. On the other hand, the importance of crossing the blood–brain barrier (BBB) for an anti-HIV drug remains unclear, since anti-HIV drugs do not reside on the brain side of the BBB and prefer to remain on the blood side. However, ineffective penetration of anti-HIV drugs into the brain can trigger HIV recurrence due to the remaining reservoirs in the central nervous system (CNS) and can cause HIV-associated neurocognitive disorders [54,55,56]. In silico pharmacokinetic studies employed for H-2, H-6, H-32, H-39, H-62, H-64, H-86, H-189, and H-201 indicated that all of them possessed moderate drug-likeness profiles with appropriate CIQPlogS and QPlogPo/w values. It was observed that their QPlogBB and CNS activity parameters were beyond the limits.

## 4. Materials and Methods

### 4.1. In Silico Docking Assessment and ADME Prediction

#### 4.1.1. Ligand Library Creation

Compounds were designed associated with our target molecule, L-HIPPO. Then, 4 design strategies were enacted on L-HIPPO to obtain more effective anti-HIV derivatives. Aromatic group-substituted L-HIPPO derivatives were designed using the proper structures from the PubChem and ZINC databases.

#### 4.1.2. Protein Preparation

The crystal structures of MA with its inhibitor IP6 were retrieved from the RSCB database (PDB IDs: 7E1I, 7E1J, and 7E1K) [44]. The raw file was prepared for the docking assessment by the PrepWizard module of Maestro. The missing chains were added automatically by Prime and the protonation state was calculated by PropKa at physiological pH. Finally, the receptor–ligand complex was minimized by optimized potential liquid simulation (OPLS_2005 (Schrödinger, LLC, New York, NY, USA, 2016)) force field.

#### 4.1.3. Docking Grid Generation

Grid generation in Maestro was used to determine the docking grid, which was centered on the crystallographic inhibitor present in the crystal structure and extended to a space of 25 × 25 × 25 angstrom. The generated grid was used for the docking experiments.

#### 4.1.4. Ligand Preparation

Compounds were sketched and cleaned in the Maestro workspace and were prepared with energy minimization using the OPLS_2005 force field at physiological pH using the LigPrep module. Then, the best minimized structures were submitted to the docking experiments without further modifications. The flexible ligand alignment tool was applied to superimpose our structurally similar ligands.

#### 4.1.5. Docking Experiments

First of all, in order to validate the docking protocol, a self-docking experiment was performed. The crystallographic inhibitor (IP6) was removed from the receptor, prepared, and minimized by the LigPrep module in Maestro using EpiK at physiological pH. The optimum structure (lowest energy) was used for the self-docking procedure. The obtained ligand was submitted to Glide/XP docking protocols (Schrödinger, LLC, NY, USA). Afterward, the same docking procedures were carried out for all designed compounds.

#### 4.1.6. In Silico ADME Studies

Some crucial pharmacokinetic properties of the new L-HIPPO derivatives were predicted by the QikProp module in Maestro (QikProp, Schrödinger, LLC, NY, 2016).

## 5. Conclusions

In the current work, we performed molecular docking studies for eight new L-HIPPO derivatives (SCH-1, SCH-2, FH-1, FH-2, USH-1, USH-2, BH-1, and BH-2) and obtained the most promising results with BH-1 and BH-2. Then, we repeated the same in silico procedures for the new BH-1 and BH-2 derivatives. Compounds H-2 and H-6 presented more significant values, followed by compounds H-32, H-39, H-62, H-64, H-86, H-189, and H-201 in a large number of derivatives. Overall, these last nine compounds with optimum ADME features could be identified as lead candidates, which are expected to show superior mechanistic antiviral effects for future studies. Our next step will be synthesis of these new L-HIPPO derivatives, measuring their in vitro MA binding affinity to support the results of the in silico studies. Our anti-HIV drug candidates might be a solution for elimination of whole HIV-infected cells, making an enormous impact globally.

## Figures and Tables

**Figure 1 pharmaceuticals-15-01255-f001:**
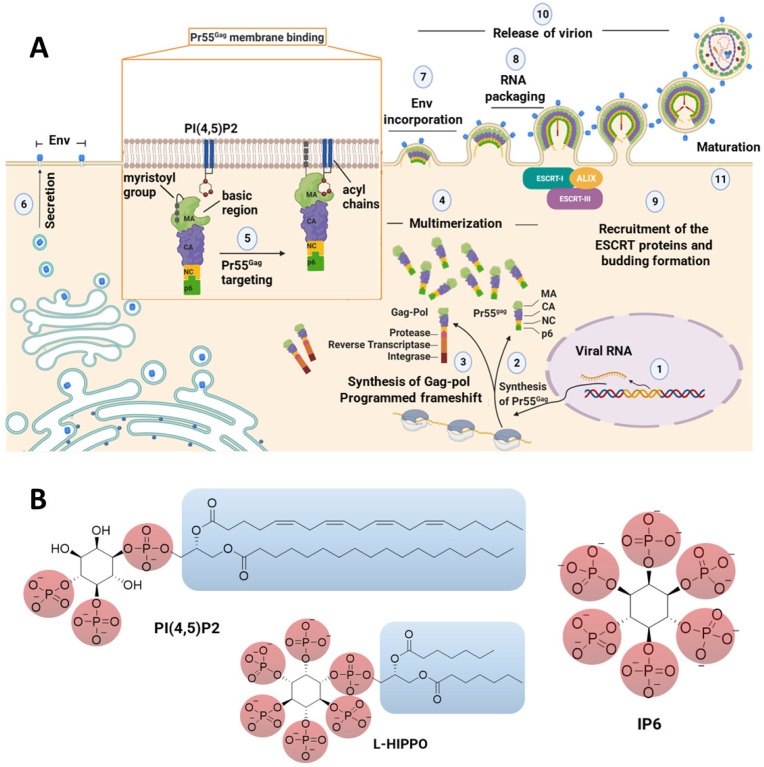
(**A**) Representation of the late stage of the HIV-1 cycle and interactions of the MA domain with PI(4,5)P2 in the membrane. (**B**) The chemical structures of PI(4,5)P2, IP6, and L-HIPPO. ALIX: ALG2-interacting protein X, CA: capsid domain, Env: viral envelope glycoprotein, ESCRT: endosomal sorting complex required for transport, MA: matrix domain, NC: nucleocapsid domain.

**Figure 2 pharmaceuticals-15-01255-f002:**
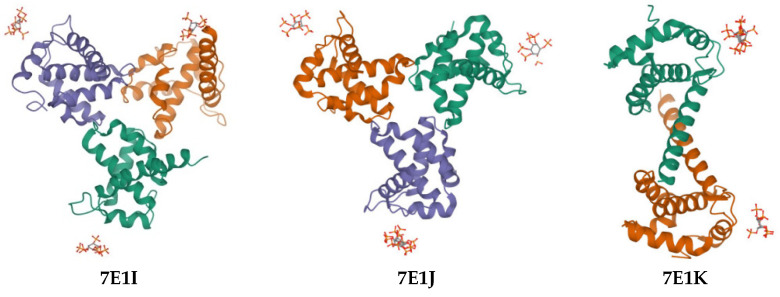
The X-ray crystallographic structures of MA, which were recently released by our research group [44].

**Figure 3 pharmaceuticals-15-01255-f003:**
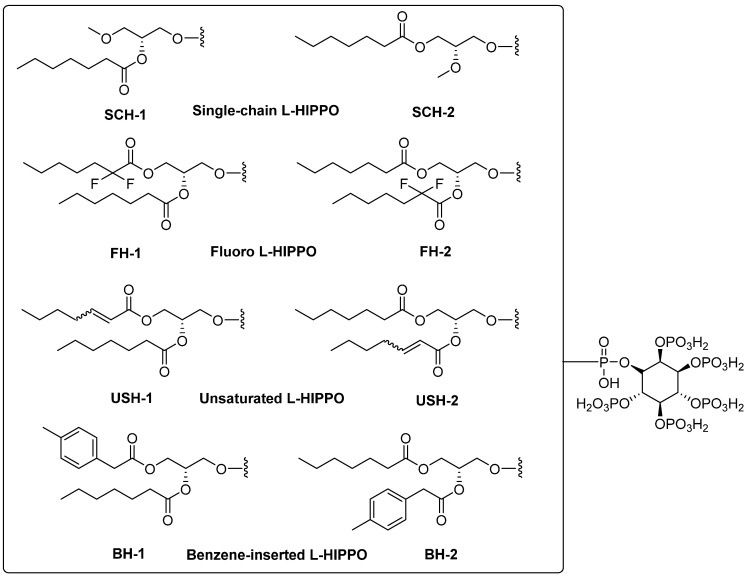
New L-HIPPO derivatives based on four new design strategies.

**Figure 4 pharmaceuticals-15-01255-f004:**
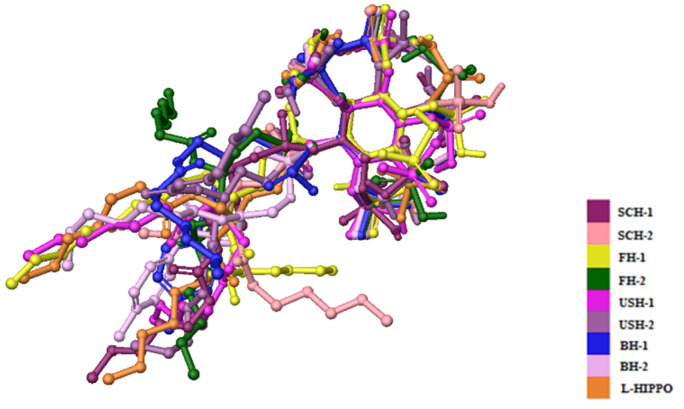
Alignment of eight new L-HIPPO derivatives (SCH-1, SCH-2, FH-1, FH-2, USH-1, USH-2, BH-1, and BH-2) in the MA domain (PDB ID: 7E1I).

**Figure 5 pharmaceuticals-15-01255-f005:**
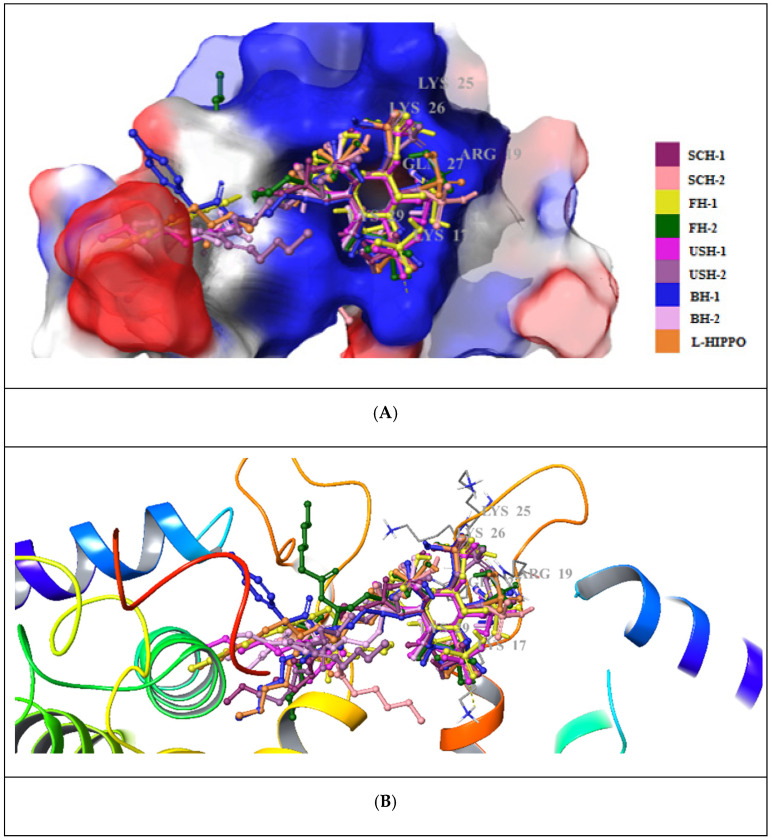
Docking poses of eight new L-HIPPO derivatives (yellow dashes: hydrogen bonding) in the MA domain (PDB ID: 7E1I) (**A**) Surface presentation. (**B**) Ribbon presentation. (**C**) Docking interactions of BH-1, BH-2, and L-HIPPO in the MA domain (PDB ID: 7E1I).

**Figure 6 pharmaceuticals-15-01255-f006:**
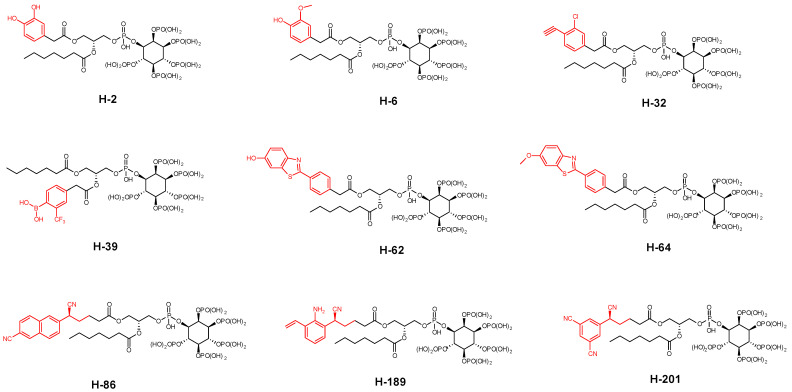
Aromatic group-based new L-HIPPO derivatives.

**Figure 7 pharmaceuticals-15-01255-f007:**
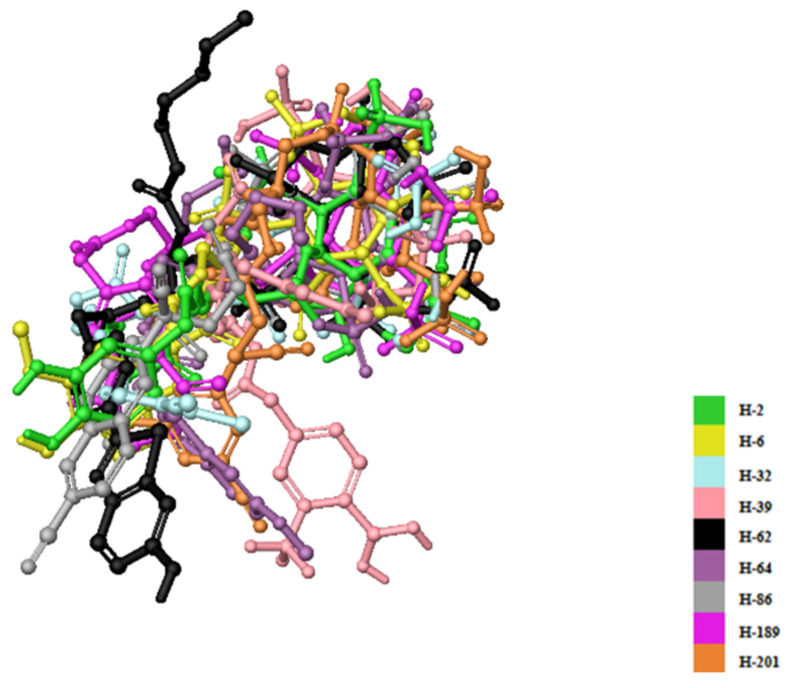
Alignment of nine new L-HIPPO derivatives (H-2, H-6, H-32, H-39, H-62, H-64, H-86, H-189, and H-201) in the MA domain (PDB ID: 7E1I).

**Figure 8 pharmaceuticals-15-01255-f008:**
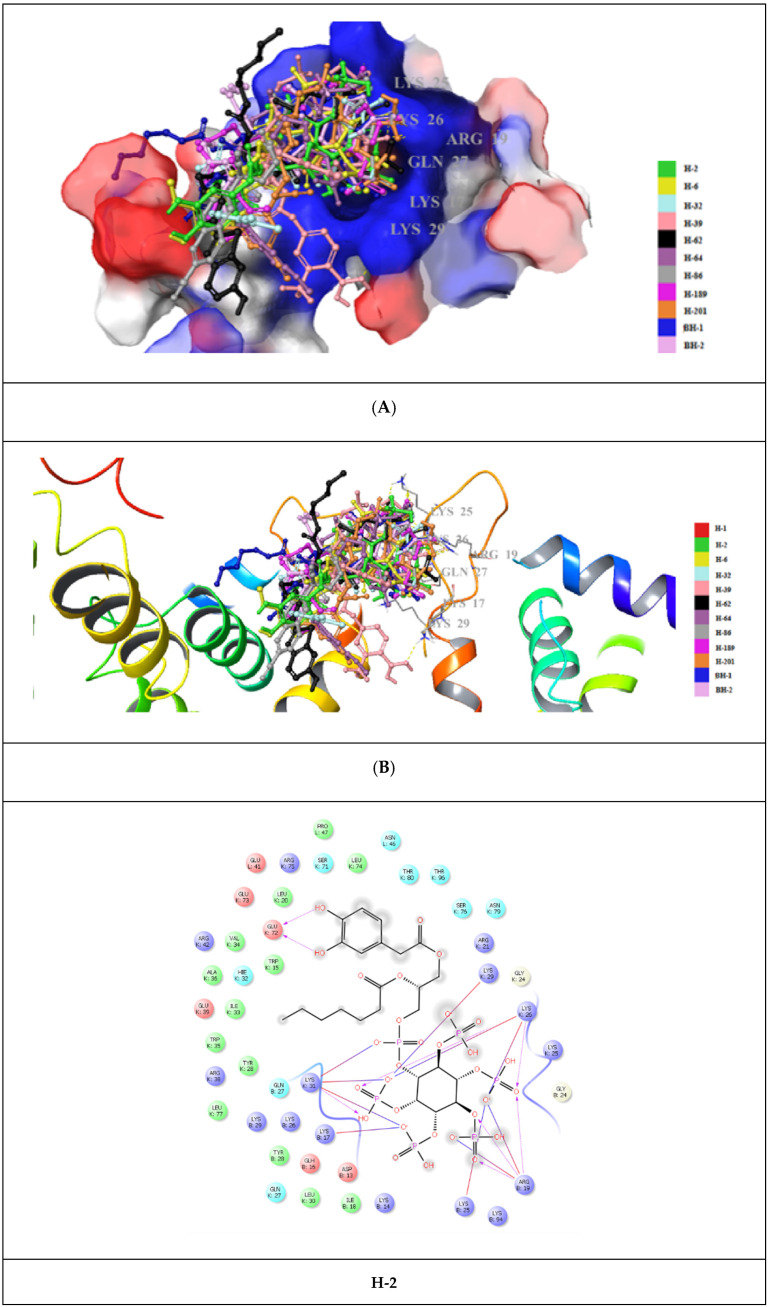
Docking poses of nine new superimposed L-HIPPO derivatives (yellow dashes: hydrogen bonding) in the MA domain (PDB ID: 7E1I). (**A**) Surface presentation. (**B**) Ribbon presentation). (**C**) Docking interactions of H-2 and H-6 in the MA domain (PDB ID: 7E1I).

**Table 1 pharmaceuticals-15-01255-t001:** Docking score (kcal/mol), glide gscore (kcal/mol), and glide emodel (kcal/mol) results of eight new L-HIPPO derivatives in the MA domain (PDB IDs: 7E1I, 7E1J, and 7E1K).

Compound	PDB IDs								
	7E1I			7E1J			7E1K		
	Docking Score	Glide Gscore	Glide Emodel	Docking Score	Glide Gscore	Glide Emodel	Docking Score	Glide Gscore	Glide Emodel
**SCH-1**	−6.018	−6.032	−188.439	−4.634	−4.649	−51.858	−5.809	−5.823	−94.442
**SCH-2**	−6.153	−6.167	−166.921	−3.943	−3.957	−46.579	−6.029	−6.043	−105.004
**FH-1**	−6.268	−6.282	−158.902	−4.407	−4.407	−53.872	−5.627	−4.483	−88.529
**FH-2**	−5.303	−5.303	−162.556	−5.777	−5.777	−74.193	−5.699	−5.712	−86.413
**USH-1**	−6.259	−6.273	−141.432	−3.770	−3.784	−32.005	−6.221	−6.235	−110.130
**USH-2**	−6.246	−6.259	−133.188	−3.660	−3.660	−35.692	−6.106	−6.119	−101.126
**BH-1**	−8.123	−8.137	−158.580	−6.253	−6.953	−40.835	−7.484	−7.497	−101.439
**BH-2**	−8.074	−8.088	−153.840	−6.070	−6.070	−41.406	−7.221	−7.235	−102.794
**IP6**	−6.914	−6.210	−162.260	−4.677	−4.692	−60.450	−5.916	−5.931	−109.656
**L-HIPPO**	−7.259	−7.273	−141.432	−5.770	−5.784	−32.005	−7.107	−7.121	−110.130

**Table 2 pharmaceuticals-15-01255-t002:** The H-bond and salt bridge formation distances of BH-1, BH-2, and L-HIPPO (angstrom) in the MA domain (PDB ID: 7E1I).

Compound	Residue	Distance	
		H-Bond	Salt-Bridge Formation
**L-HIPPO**	Arg19	1.83 and 1.61	4.31
	Lys25		4.20
	Lys26	2.07	4.67
	Lys29	1.96	
	Lys31	2.23	3.65
**BH-1**	Arg19	1.57 and 1.93	4.20
	Lys25		2.76
	Lys26	2.16	
	Lys29	2.41	
	Lys31	2.40 and 2.53	
**BH-2**	Arg19	2.02	
	Lys25		2.88
	Lys26	2.54	4.97
	Lys29	2.62	4.72
	Lys31	1.94 and 1.77	

**Table 3 pharmaceuticals-15-01255-t003:** Docking score (kcal/mol), glide gscore (kcal/mol), and glide emodel (kcal/mol) results of aromatic group-based nine new L-HIPPO derivatives in the MA domain (PDB IDs: 7E1I, 7E1J, and 7E1K).

Compound	PDB IDs								
	7E1I			7E1J			7E1K		
	Docking Score	Glide Gscore	Glide Emodel	Docking Score	Glide Gscore	Glide Emodel	Docking Score	Glide Gscore	Glide Emodel
**H-2**	−11.587	−11.564	−190.864	−9.236	−9.175	−40.375	−10.587	−10.432	−120.138
**H-6**	−11.615	−11.629	−192.714	−9.675	−9.689	−84.165	−10.777	−10.790	−96.017
**H-32**	−11.023	−11.037	−192.043	−9.424	−9.438	−38.360	−10.140	−10.153	−107.200
**H-39**	−11.341	−11.355	−185.992	−9.432	−9.466	−44.267	−10.213	−10.227	−94.803
**H-62**	−11.275	−11.290	−191.833	−9.289	−9.303	−54.908	−10.314	−10.328	−162.156
**H-64**	−11.110	−11.124	−174.378	−9.213	−9.227	−51.333	−10.715	−10.729	−126.444
**H-86**	−11.298	−11.312	−177.756	−9.033	−9.047	−77.899	−10.249	−10.262	−101.827
**H-189**	−11.498	−11.452	−180.730	−9.625	−9.639	−82.109	−10.373	−10.387	−106.364
**H-201**	−11.403	−11.417	−186.951	−9.020	−9.034	−38.228	−10.348	−10.962	−117.939

**Table 4 pharmaceuticals-15-01255-t004:** Predicted ADME properties of new L-HIPPO derivatives.

Compound	CIQPlogS*	QPlogBB*	CNS*	QPlogPo/w*	Rule of Five*	Rule of Three*
**H-2**	−4.571	−7.647	−2	−0.425	3	1
**H-6**	−4.910	−9.260	−2	−0.496	3	1
**H-32**	−5.752	−8.159	−2	1.176	3	1
**H-39**	−6.432	−10.160	−2	1.016	3	1
**H-62**	−6.072	−8.966	−2	1.112	3	1
**H-64**	−5.639	−9.209	−2	0.229	3	1
**H-86**	−6.173	−10.299	−2	0.223	3	1
**H-189**	−4.966	−8.993	−2	−0.100	3	1
**H-201**	−5.885	−9.326	−2	−1.441	3	1

## Data Availability

Data are contained within the article.

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
