# Peer review of "Identification of New L-Heptanoylphosphatidyl Inositol Pentakisphosphate Derivatives Targeting the Interaction with HIV-1 Gag by Molecular Modelling Studies"

_pharmaceuticals, 2022, doi:10.3390/ph15101255_

Round 1

Reviewer 2 Report

The work presented to me for review concerns the description of the interaction of new  L-Heptanoylphosphatidyl Inositol Pen- 2 takisphosphate derivatives with HIV-1 Gag using molecular modeling. The authors first describe the results for 8 derivatives, and then on their basis create a library containing up to 213 compounds. The work is prepared carefully, reliably, and the results are properly described. The description of the experiment does not raise my reservations either. I believe that the publication is innovative. I only have a few minor comments:

- I suggest increasing the font size in Figure 1A as it is now hard to read.

- Line 134 - the reference to Figure 1 should not be written in red

- Figure 3 is too small compared to the rest of the text. Should be edited to make it more readable.
